# An introduction to learning algorithms and potential applications in geomorphometry and earth surface dynamics

**Andrew Valentine[1] and Lara Kalnins[2]**

[1]Department of Earth Sciences, Universiteit Utrecht, Postbus 80.021, 3508TA Utrecht, The Netherlands.
[2]Department of Earth Sciences, Science Labs, Durham University, Durham, DH1 3LE, UK.

*Correspondence to:* Andrew Valentine (a.p.valentine@uu.nl)

**Abstract.** 'Learning algorithms' are a class of computational tool designed to infer information from a dataset, and then apply that information predictively. They are particularly well-suited to complex pattern recognition, or to situations where a mathematical relationship needs to be modelled, but where the underlying processes are not well-understood, are too expensive to compute, or where signals are over-printed by other effects. If a representative set of examples of the relationship can be constructed, a learning algorithm can assimilate its behaviour, and may then serve as an efficient, approximate computational implementation thereof. A wide range of applications in geomorphometry and earth surface dynamics may be envisaged, ranging from classification of landforms through to prediction of erosion characteristics given input forces. Here, we provide a practical overview of the various approaches that lie within this general framework, review existing uses in geomorphology and related applications, and discuss some of the factors that determine whether a learning algorithm approach is suited to any given problem.

## 1 Introduction

The human brain has a remarkable capability for identifying patterns in complex, noisy datasets, and then applying this knowledge to solve problems or negotiate new situations. The research field of 'learning algorithms' (or 'machine learning') centres around attempts to replicate this ability via computational means, and is a cornerstone of efforts to create 'artificial intelligence'. The fruits of this work may be seen in many different spheres—learning algorithms feature in everything from smartphones to financial trading. As we shall discuss in this paper, they can also prove useful in scientific research, providing a route to tackling problems that are not readily solved by conventional approaches. We will focus particularly on applications falling within geomorphometry and earth surface dynamics, although the fundamental concepts are applicable throughout the geosciences, and beyond.

This paper does not attempt to be comprehensive. It is impossible to list every problem that could potentially be tackled using a learning algorithm, or to describe every technique that might somehow be said to involve 'learning'. Instead, we aim to provide a broad overview of the possibilities and limitations associated with these approaches. We also hope to highlight some of the issues that ought to be considered when deciding whether to approach a particular research question by exploring the use of learning algorithms.

The artificial intelligence literature is vast, and can be confusing. The field sits at the interface of computer science, engineering, and statistics: each brings its own perspective, and sometimes uses different language to describe essentially the same concept. A good starting point is the book by Mackay (2003), although this assumes a certain level of mathematical fluency; Bishop (2006) is drier and has a somewhat narrower focus, but is otherwise aimed at a similar readership. Unfortunately, the nature of the field is such that there are few good-quality reviews targeted to the less mathematically-inclined, although one example can be found in Olden et al. (2008). There is also a wealth of tutorials and other course material available online, varying in scope and quality. Finally, we draw readers' attention to a recent review by Jor-

dan and Mitchell (2015), offering a broad survey of machine learning and its potential applications.

Learning algorithms are computational tools, and a number of software libraries are available which provide users with a relatively straightforward route to solving practical problems. Notable examples include `pybrain` and `scikit-learn` for Python (Schaul et al., 2010; Pedregosa et al., 2011), and the commercially-available 'Statistics and Machine Learning' and 'Neural Networks' toolboxes for Matlab. Most major techniques are also available as packages within the statistical programming language, R. Nevertheless, we encourage readers with appropriate interest and skills to spend some time writing their own implementations for basic algorithms: our experience suggests that this is a powerful aid to understanding how methods behave, and provides an appreciation for potential obstacles or pitfalls. In principle, this can be achieved using any mainstream programming language, although it is likely to be easiest in a high-level language with built-in linear algebra support (e.g. Matlab, Python with `NumPy`). Efficient 'production' applications are likely to benefit from use of the feature-intensive and highly-optimised tools available within the specialist software libraries mentioned above.

This paper begins with a brief overview of the general framework within which learning algorithms operate. We then introduce three fundamental classes of problem that are often encountered in geomorphological research, and which seem particularly suited to machine learning solutions. Motivated by this, we survey some of the major techniques within the field, highlighting some existing applications in geomorphology and related fields. Finally, some of the practical considerations that affect implementation of these techniques are discussed, and we highlight some issues that should be noted when considering exploring learning algorithms further.

## 1.1   Learning algorithms: A general overview

Fundamentally, a learning algorithm is a set of rules that are designed to find and exploit patterns in a dataset. This is a familiar process when patterns are known (or assumed) to take a certain form—consider fitting a straight line to a set of data points, for example—but the power of most learning algorithm approaches lies in their ability to handle complex, arbitrary structures in data. Traditionally, a distinction is drawn between 'supervised' learning algorithms, which are aimed at training the system to recognise known patterns, features, or classes of object; and 'unsupervised' learning, aimed finding patterns in the data that have not previously been identified or that are not well defined. Supervised learning typically involves optimising some pre-defined measure of the algorithm's performance—perhaps minimising the difference between observed values of a quantity and those predicted by the algorithm—while in unsupervised learning, the goal is usually for the system to reach a mathematically-stable state.

At a basic level, most learning algorithms can be regarded as 'black boxes': they take data in, and then output some quantity based upon that data. The detail of the relationship between inputs and outputs is governed by a number of adjustable parameters, and the 'learning' process involves tuning these to yield the desired performance. Thus, a learning algorithm typically operates in two modes: a learning or 'training' phase, where internal parameters are iteratively updated based on some 'training data', and an 'operational' mode in which these parameters are held constant, and the algorithm outputs results based on whatever it has learned. Depending on the application, and the type of learning algorithm, training may operate as 'batch learning'—where the entire dataset is assimilated in a single operation—or as 'online learning', where the algorithm is shown individual data examples sequentially and updates its model parameters each time. This may be particularly useful in situations where data collection is ongoing, and it is therefore desirable to be able to refine the operation of the system based on this new information.

In the context of learning algorithms, a 'dataset' is generally said to consist of numerous 'data vectors'. For our purposes, each data vector within the dataset will usually correspond to the same set of physical observations made at different places in space or time. Thus, a dataset might consist of numerous stream profiles, or different regions extracted from a LiDAR-derived digital elevation model (DEM). It is possible to combine multiple, diverse physical observations into a single data vector: for example, it might be desirable to measure both the cross-section and variations in flow rate across streams, and regard both as part of the same data vector. It is important to ensure that all data vectors constituting a given dataset are obtained and processed in a similar manner, so that any difference between examples can be attributed solely to physical factors. In practice, pre-processing and 'standardising' data to enhance features that are likely to prove 'useful' for the desired task can also significantly impact performance: we return to this in Section 4.1.

## 2   Some general classes of geomorphological problem

Broadly speaking, we see three classes of problem where learning algorithms can be particularly useful in geomorphology: classification and cataloguing; cluster analysis and dimension reduction; and regression and interpolation. All represent tasks that can be difficult to implement effectively via conventional means, and which are fundamentally data-driven. However, there can be considerable overlap between all three, and many applications will not fit neatly into one category.

## 2.1 Classification and cataloguing

Classification problems are commonplace in observational science, and provide the canonical application for supervised learning algorithms. In the simplest case, we have a large collection of observations of the same type—perhaps cross-sections across valleys—and we wish to assign each to one of a small number of categories (for example, as being of glacial or riverine form). In general, this kind of task is straightforward to an experienced human eye. However, it may be difficult to codify the precise factors that the human takes into account, preventing their implementation as computer code: simple rules break down in the face of the complexities inherent to real data from the natural world. With a learning algorithm approach, the user typically classifies a representative set of examples by hand, so that each data vector is associated with a 'flag' denoting the desired classification. The learning algorithm then assimilates information about the connection between observations and classification, seeking to replicate the user's choices as closely as possible. Once this training procedure has been completed, the system can be used operationally to classify new examples, in principle without the need for further human involvement.

Beyond an obvious role as a labour-saving device, automated systems may enable users to explore how particular factors affect classification. It is straightforward to alter aspects of data processing, or the labelling of training examples, and then re-run the classification across a large dataset. Another advantage lies in the repeatable nature of the classification: observations for a new area, or obtained at a later date, can be processed in exactly the same manner as the original dataset, even if personnel differ. It is also possible to use multiple datasets simultaneously when performing the classification—for example, identification of certain topographic features may be aided by utilising high-resolution local imagery, plus lower-resolution data showing the surrounding region, or land use classification may benefit from using topography together with satellite imagery.

It is sometimes claimed that it is possible to somehow 'interrogate' the learning algorithm, to discover its internal state and understand which aspects of the data are used to make a particular classification. This information could offer new insights into the physical processes underpinning a given problem. For the simplest classifiers, this may be possible, but in general we believe it ought to be approached with some scepticism. Classification systems are complex, and subtle interactions between their constituent parts can prove important. Thus, simplistic analysis may prove misleading. A more robust approach, where feasible, would involve classifying synthetic (artificial) data, and exploring how parameters controlling the generation of this affect results (see also Hillier et al., 2015). It may also be instructive to explore how performance varies when different subsets of observables are used.

Conventionally, classification problems assume that all examples presented to the system can be assigned to one category or another. A closely-related problem, which we choose to call 'cataloguing', involves searching a large dataset for examples of a particular feature—for example, locating moraines or faults in regional-scale topographic data. This introduces additional challenges: each occurrence of the feature should be detected only once, and areas of the dataset that do not contain the desired feature may nevertheless vary considerably in their characteristics. As a result, cataloguing problems may require an approach that differs from other classification schemes.

Examples of classification problems in geomorphology where machine learning techniques have been applied include classifying elements of urban environments (Miliaresis and Kokkas, 2007), river channel morphologies (Beechie and Imaki, 2014), and landslide susceptibility levels (e.g. Brenning, 2005). An example of a cataloguing problem is given in Valentine et al. (2013), aimed at identifying seamounts in a range of tectonic settings.

## 2.2 Cluster analysis and dimension reduction

Classification problems arise when the user has prior knowledge of the features they wish to identify within a given dataset. However, in many cases we may not fully understand how a given process manifests itself in observable phenomena. Cluster analysis and dimension reduction techniques provide tools for 'discovering' structure within datasets, by identifying features that frequently occur, and by finding ways to partition a dataset into two or more parts, each with a particular character. An accessible overview of the conceptual basis for cluster analysis, as well as a survey of available approaches, can be found in Jain (2010).

In many applications, data vectors are overparameterised: the representations used for observable phenomena have more degrees of freedom than the underlying physical system. For example, local topography might be represented as a grid of terrain heights. If samples are taken every ten metres, then eleven samples span a distance of $100\,\mathrm{m}$, and a $100\,\mathrm{m} \times 100\,\mathrm{m}$ area is represented by 121 distinct measurements. A single 121-dimensional data vector is obtained by 'unwrapping' the grid according to some well-defined scheme—perhaps by traversing lines of latitude. Nevertheless, the underlying process of interest might be largely governed by a handful of parameters. This implies that there is a high level of redundancy within the data vector, and it could therefore be transformed to a lower-dimensional state without significant loss of information. This may be beneficial, either in its own right or as an adjunct to other operations: low dimensional systems tend to be easier to handle computationally, and similarities or differences between examples may be clearer after transformation. Dimension reduction algorithms aim to find the optimal low-dimension representation of a given dataset.

One particularly important application of dimension reduction lies in visualisation. Where individual data vec-

tors are high-dimensional, it may be difficult to devise effective means of plotting them in (typically) two dimensions. This makes it difficult to appreciate the structure of a dataset, and how different examples relate to one another. In order to tackle this problem, learning algorithms may be used to identify a two-dimensional representation of the dataset that somehow preserves the higher-dimensional relationships between examples. This process involves identifying correlations and similarities between individual data vectors, and generally does not explicitly incorporate knowledge of the underlying physical processes. Thus, the coordinates of each example within the low-dimensional space may not have any particular physical significance, but examples that share common characteristics will yield closely-spaced points. Thus, it may be possible to identify visual patterns, and hence discover relationships within the high-dimensional data.

Geomorphological applications of cluster analysis include identifying flow directions from glacial landscapes (Smith et al., 2016), identifying different types of vegetation (Belluco et al., 2006), or extracting common structural orientations from a laser scan of a landslide scarp (Dunning et al., 2009). An example of dimension reduction comes again from landslide susceptibility studies, this time aimed at identifying the most influential observables within a suite of possibilities (Baeza and Corominas, 2001).

## 2.3 Regression and interpolation

The third class of problem involves learning relationships between physical parameters, in order to make predictions or to infer properties. Very often, it is known that one set of observable phenomena are closely-related to a different set— but the details of that relationship may be unknown, or it may be too complex to model directly. However, if it is possible to obtain sufficient observations where both sets of phenomena have been measured, a learning algorithm can be used to represent the link, and predict one given the other—for example, an algorithm might take measurements of soil properties and local topography, and output information about expected surface run-off rates. Alternatively, the same training data could be used to construct a system that infers the soil parameters given topography and run-off measurements. This may be useful when there are fewer measurements available for one of the physical parameters, perhaps because it is harder or more expensive to measure: once trained on examples where this parameter has been measured, the algorithm can be used to estimate its value in other locations based on the more widely available parameters.

Questions of this sort may be framed deterministically— so that the system provides a single prediction—or statistically, where the solution is presented as a probability distribution describing the range of possible outcomes. The choice of approach will depend upon the nature of the underlying problem, and upon the desired use of the results. In general, probabilistic approaches are desirable, since they provide a more realistic characterisation of the system under consideration—deterministic approaches can be misleading when more than one solution is compatible with available data, or where uncertainties are large. However, in some cases it may be difficult to interpret and use information presented as a probability distribution. For completeness, we observe that most learning algorithms have their roots in statistical theory, and even when used 'deterministically', the result is formally defined within a statistical framework.

In geomorphology, a common application of machine learning for regression and interpolation is to link widely available remote sensing measurements with underlying parameters of interest that cannot be measured directly: for example, sediment and chlorophyll content of water from colour measurements (Krasnopolsky and Schiller, 2003) or marine sediment properties from bathymetry and proximity to the coast (Li et al., 2011; Martin et al., 2015).

## 3 Some popular techniques

The aforementioned problems can be tackled in almost any number of ways: there is rarely a single 'correct' approach to applying learning algorithms to any given question. As will become clear, once a general technique has been selected, there remains a considerable array of choices to be made regarding its precise implementation. Usually, there is no clear reason to make one decision instead of another—often, literature describes some 'rule of thumb', but its underlying rationale may not always be obvious. A certain amount of trial-and-error is generally required to obtain optimal results with a learning algorithm. This should perhaps be borne in mind when comparisons are drawn between different approaches: although many studies can be found in the literature that conclude that one method outperforms another for a given problem, it is unlikely that this has been demonstrated to hold for all possible implementations of the two methods. It is also worth noting that the relationship between performance and computational demands may differ between algorithms: a method that gave inadequate performance on a desktop computer a decade ago may nevertheless excel given the vastly-increased resources of a modern, high-performance machine.

In what follows, we outline a selection of common methods, with an emphasis on conveying general principles rather than providing precise formal definitions. There is no particular rationale underpinning the methods we choose to include here, beyond a desire to cover a spectrum of important approaches. Other authors would undoubtedly make a different selection (for example, see Wu et al. (2008), although this has a narrower scope than the present work). In an effort to promote readability, we order our discussion roughly according to complexity, although this is not an objective assessment. A brief summary of the methods discussed can be found in Table 1.

| Method | Generally used for: | | | | Learning mode | | |
| | Classification | Clustering | Dimension reduction | Regression | Supervised | Unsupervised | Deterministic |
|---|---|---|---|---|---|---|---|
| Decision trees | ✓ | | | | ✓ | | |
| K-Means | | ✓ | | | | ✓ | |
| PCA | | | ✓ | | | ✓ | ✓ |
| Neural networks | ✓ | | ✓ | ✓ | ✓ | ✓ | |
| SVMs | ✓ | | | | | ✓ | |
| SOMs | | ✓ | ✓ | | | ✓ | |
| Bayesian inference | ✓ | ✓ | | ✓ | ✓ | ✓ | (✓) |

**Table 1.** Summary of methods discussed in this paper. For each of the 'popular techniques' discussed in Section 3, we indicate the classes of problem for which they are generally used (as described in Section 2); whether the method generally operates as 'supervised' learning (based on optimising a pre-determined performance measure), or unsupervised (attempting to reach a stable state); and whether the method is typically 'deterministic', so that it is guaranteed to yield identical results each time it is applied to a given dataset. Note that Bayesian inference is itself deterministic, but it is most often encountered in contexts where it is applied to randomly-chosen observations. The indications given here are not intended to be exhaustive: it may be possible to adapt each technique to suit the full range of applications.

## 3.1 Decision trees and Random Forests

A decision tree is a system that takes a data vector, and processes it via a sequence of `if-then-else` constructs ('branch points') until an output state can be determined (see Fig. 1). This is clearly well-suited to addressing simple classification problems, and to predicting the states of certain physical systems: essentially, the system resembles a flowchart. In this context, 'learning' involves choosing how the dataset should be partitioned at each branch.

Typically, each data vector contains a number of 'elements'—distinct observations, perhaps made at different points in space or time, or of different quantities relevant to the phenomenon of interest. Each vector is also associated with a particular 'desired outcome'—the classification or state that the tree should output when given that example. Basic decision tree generation algorithms aim to identify a test that can be applied to any one element, which separates desired outcomes as cleanly as possible (e.g. Fig. 1a,b). This is typically quantified using a measure such as 'information entropy', which assesses the degree to which a system behaves predictably. Once the training data has been partitioned into two sets, the algorithm can be applied recursively on each, until a complete tree has been constructed. Commonly-encountered tree generation schemes include `ID3` (Quinlan, 1986) and its successor `C4.5` (Quinlan, 1993).

Tree generation assumes that the data are perfect, and will therefore continue adding branch points until all training data can be classified as desired. When real-world datasets—which invariably contain errors and 'noise'—are used, this tends to result in overly complex trees, with many branches. This phenomenon is known as 'overfitting', and tends to result in a tree with poor generalisation performance: when used to process previously unseen examples, the system does not give the desired outcome as often as one might hope. It is therefore usual to adopt some sort of 'pruning' strategy, by which certain branches are merged or removed. Essentially, this entails prioritising simple trees over perfect performance for training data; a variety of techniques exist, and the choice will probably be application-specific.

Another approach to this issue, and to the fact that in many problems the number of data elements vastly exceeds the number of possible outcomes, lies in the use of 'random forests' (Breiman, 2001). By selecting data vectors from the training set at random (with replacement), and discarding some randomly-chosen elements from these, we can construct a number of new datasets. It is then straightforward to build a decision tree for each of these. Typically, this results in trees that perform well for some—but not all—examples. However, if we use each tree to predict an outcome for a given data vector, and then somehow compute the average of these predictions, performance is usually significantly better than can be achieved with any one tree. This strategy, where a number of similar, randomised systems are constructed and then used simultaneously, is sometimes referred to as an 'ensemble' method. Again, when treated in more detail, a variety of approaches are possible, and the intended application may help dictate which should be used.

A recent example of an application of random forests to Earth surface data can be found in Martin et al. (2015). Here, the goal is to predict the porosity of sediments lying on the ocean floor from a range of different data, including measures such as water depth or distance from sediment-producing features. Such observations are much easier to obtain than direct measurement of the seafloor, which obviously requires ocean-bottom sampling. Using training data from locations where such samples have been collected, a random forest is constructed that enables porosity predictions to be made throughout the oceans. This concept could readily be adapted to a variety of other situations where local physical properties must be estimated from remotely-sensed data. Further discussion of the use of random forests for interpola-

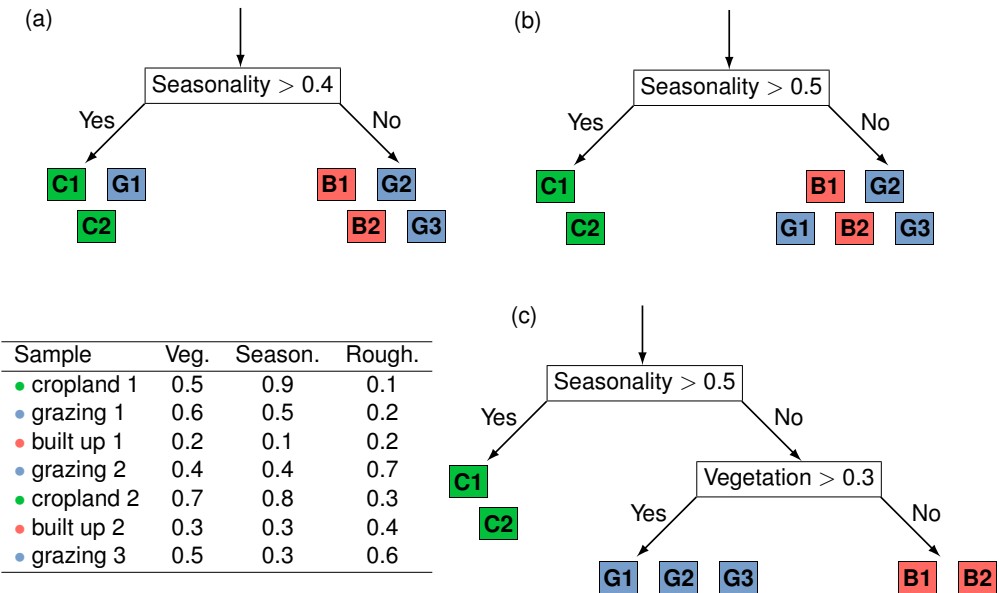

**Figure 1.** Evolution of a decision tree for land use data based on normalised parameters for vegetation index, seasonal colour variability, and topographic roughness. (a) The algorithm tests a variety of conditions for the first branch point, looking for a condition such as (b) that cleanly separates different classes. (c) A second branch point is added to differentiate between built up and grazing land.

tion between samples can be found in, for example, Li et al. (2011), who test a variety of techniques including random forests and support vector machines in order to produce a regional map of mud content in sediments based on discrete sampling. Another example comes from Bhattacharya et al. (2007), who construct models of sediment transport using a variant of decision trees where each 'leaf' is a linear regression function, rather than a single classification; the decision tree is essentially used to choose which mathematical model to apply in each particular combination of circumstances.

## 3.2    The K-Means Algorithm

By far the most well-known technique for cluster analysis, K-means is usually said to have its origins in work eventually published as Lloyd (1982), but disseminated earlier (thus, for example, Hartigan and Wong (1979) sets out a specific implementation). The algorithm is designed to divide a set of $N$ data vectors, each consisting of $M$ elements, into $K$ clusters. These clusters are defined so that the distance of each point in the cluster from its centre is as small as possible.

The algorithm is readily understood, and is illustrated in Fig. 2. We begin by generating $K$ points, which represent our initial guesses for the location of the centre of each cluster—these may be chosen completely at random, or based on various heuristics which attempt to identify a 'sensible' initial configuration. We then assign each element of our training data to the cluster with the nearest centre (Fig. 2b). Once this has been done, we recompute the position of the central point, by averaging the locations of all points in the clus-

ter (Fig. 2c). This process of assignment and averaging is repeated (Fig. 2d) until a stable configuration is obtained (Fig. 2e). The resulting clusters may then be inspected to ascertain their similarities and differences, and new data can be classified by computing its distance from each cluster centre.

In order to implement this, it is necessary to define what the word 'distance' means in the context of comparing any two data vectors. There are a number of possible definitions, but it is most common to use the 'Euclidean' distance: the sum of the squared difference between each element of the two vectors. Thus, if $\mathbf{x}$ is a vector with $M$ elements $(x_1, x_2, \ldots, x_M)$ and $\mathbf{y}$ a second vector $(y_1, y_2, \ldots, y_M)$, then the Euclidean distance between them is defined

$$d = \sum_{i=1}^{M}(x_i - y_i)^2. \tag{1}$$

This definition is a natural extension of our everyday understanding of the concept of 'distance'. However, where data vectors are comprised of more than one class of observation—perhaps combining topographic heights with soil properties—problems can arise if the measurements differ considerably in typical scale. The Euclidean distance between two two-element data vectors $(1, 10^{-9})$ and $(1, 10^{-5})$ is very small, despite the second elements differing by four orders of magnitude, because both are negligible in comparison to the first element. It may be necessary to rescale the various measurements that make up a data vector, to ensure that they have a similar magnitude and dynamic range. Alternatively, and equivalently, the definition of 'distance' can be adapted to assign different weights to various data types.

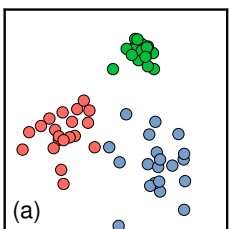 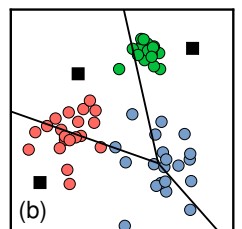 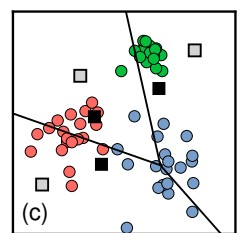 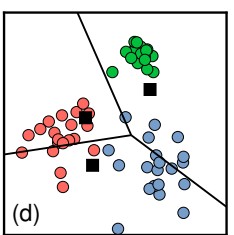 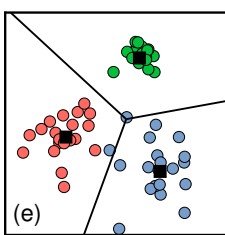

**Figure 2.** The K-Means algorithm. (a) A dataset contains points clustered around three distinct locations (colour-coded for ease of reference). (b) We first guess locations for these centres at random (black squares) and assign each datum to the nearest cluster (shown divided by black lines). (c) We then re-compute the location of each cluster centre by averaging all points within the cluster (old centres, grey squares; new centres, black squares), and (d) update the cluster assignment of each point to reflect this. It may be necessary to repeat steps (c) and (d) for several iterations until a stable partitioning of the dataset is found (e).

In its basic form, the K-means algorithm requires the user to specify the number of clusters to be sought as *a priori* information. In many cases, this may not be known, and a range of different solutions have been proposed—see, for example, Jain (2010). Fundamentally, these entail balancing increased complexity (i.e., an increased number of clusters) against any resulting reduction in the average distance between samples and the cluster centre. In general, this reduction becomes insignificant once we pass a certain number of clusters, and this is taken to provide an appropriate description of the data.

Clustering algorithms such as K-means have seen use in geomorphology as an aid to analysis or interpretation of a variety of datasets. It is common to first apply some transformation or pre-processing to raw data, to improve sensitivity to particular classes of feature. For example, Miliaresis and Kokkas (2007) take LiDAR DEMs, apply various filters designed to enhance the visibility of the built environment within the image, and then use K-means to distinguish different areas of urban environment (such as differentiating between vegetation, buildings, and roads/pavements). Similarly, Belluco et al. (2006) use the technique to assist in mapping vegetation based on remotely-sensed spectral imaging, although they find that which method performs best depends on the type of imaging and field data. On a much smaller scale, Dunning et al. (2009) use K-means and other clustering algorithms to extract discontinuity orientations from laser-derived observations of landslide scarps, which help constrain the mechanism behind the slope failure.

### 3.3 Principal component analysis

Often, the different observations comprising a given data vector are correlated, and thus not fully independent: for example, topographic heights at adjacent sites are likely to be reasonably similar to one another, and an imprint of topography will often be found in other datasets such as soil thickness or temperature. For analysis purposes, it is often desirable to identify the patterns common to multiple data elements, and to transform observations into a form where each parameter, or component, is uncorrelated from the others. Principal component analysis (PCA) provides one of the

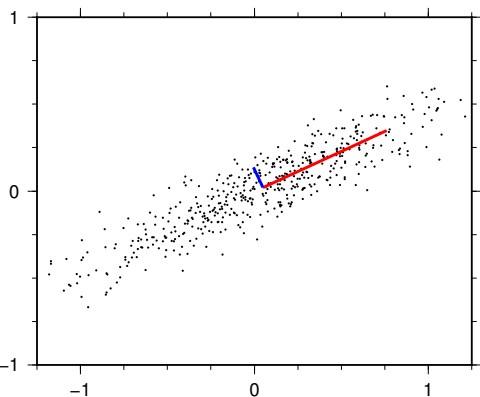

**Figure 3.** Principal component analysis of a simple dataset. Dominant principal component shown in red; secondary principal component shown in blue. Line lengths are proportional to the weight attached to each component. It is apparent that the principal components align with the directions in which the dataset shows most variance.

most common techniques for doing so, and has its roots in the work of Pearson (1901). Essentially the same mathematical operation arises in a variety of other contexts, and has acquired a different name in each: for example, 'singular value decomposition', 'eigenvector analysis', and the concept of 'empirical orthogonal functions' are all closely-related to PCA.

Numerical algorithms for performing PCA are complex, and there is usually little need for the end-user to understand their intricacies. In general terms, PCA involves finding the 'direction' in which the elements of a dataset exhibit the greatest variation, and then repeating this with the constraint that each successive direction considered must be at right angles (orthogonal) to those already found. Although easiest to visualise in two or three dimensions (see Fig. 3), the principle works in the same way for data vectors with more elements: for a set of data vectors with $M$ elements, it is possible to construct up to $M$ perpendicular directions.

Thus, the outcome of PCA is an set of orthogonal directions (referred to as principal components) ordered by their

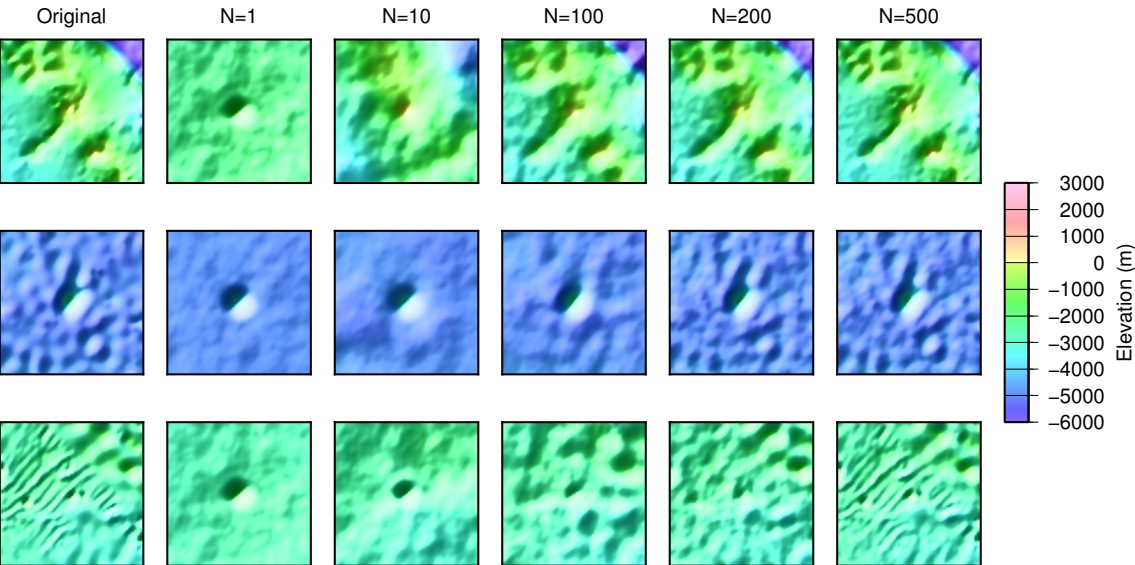

**Figure 4.** Image reconstruction using principal components. PCA has been performed on a dataset containing 1000 square 'patches' of bathymetric data, each representing an area of dimension $150 \times 150$ km, centred upon a seamount (the 'training set' used in Valentine et al., 2013). Each patch is comprised of $64 \times 64$ samples—thus, each can be seen as a 4096-dimensional object. Three examples from this set are shown here (one per row). In the left-most column we show the original bathymetry for each; then, we present reconstructions of this bathymetry using only the $N$ most significant principal components, for $N = 1, 10, 100, 200$ & $500$. It is apparent that the large-scale structures within this dataset can be represented using only 100–200 dimensions, while around 500 dimensions are required to allow some of the fine-scale structure to be represented, particularly in the third example. This still represents almost an order of magnitude reduction, in comparison to the original, 4096-dimensional, data.

importance in explaining a given dataset: in a certain sense, this can be regarded as a new set of co-ordinate axes against which data examples may be measured. The principal components may be regarded as a set of weighted averages of different combinations of the original parameters, chosen to best describe the data. Often, much of the structure of a dataset can be expressed using only the first few principal components, and PCA can therefore be used as a form of dimensionality-reduction (see Fig. 4). In a similar vein, it can form a useful precursor to other forms of analysis, such as clustering, by virtue of its role in unpicking the various signals contributing to a dataset: often, each component will exhibit sensitivity to particular physical parameters. However, particularly where the original data is composed of a variety of different physical data types, the results of PCA may not be straightforward to interpret: each principal component may be influenced by a number of disparate measurements.

One example of this can be found in Cuadrado and Perillo (1997), where PCA is performed on a dataset consisting of bathymetric measurements for a given region repeated over several months. The first principal component is then found to describe the mean bathymetry of the period, while the second provides information about the general trend of change in bathymetry over time, highlighting areas of deposition and erosion. Another typical application occurs in Baeza and Corominas (2001), where PCA is used to identify the observable parameters that best serve as predictors

of landslide hazard. A more recent example is Tamene et al. (2006), who use PCA to identify which observable parameters best explain variability in sediment yield between different river catchments in Ethiopia. This example also shows how a single principal component may be a combination of observables: their first component, which explains about 50% of the variability, is dominated by a combination of topographic variables, such as height difference, elongation ratio, and catchment area. The second component, which explains approximately 20% of the variability, is dominated by variables associated with lithology and land use/vegetation cover.

### 3.4   Neural Networks

Perhaps the most varied and versatile class of algorithm discussed in this paper is the neural network. As their name suggests, these were originally developed as a model for the workings of the brain, and they can be applied to tackling a wide range of problems. It has been shown (e.g. Hornik, 1991) that neural networks can, in principle, be used to represent arbitrarily complex mathematical functions, and their use is widespread in modern technology—with applications including tasks such as voice recognition or automatic language translation. They may be used to tackle problems falling in all three of the classes discussed in Section 2. A comprehensive introduction to neural networks may be

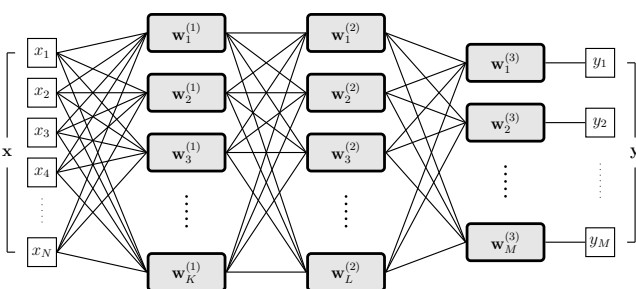

**Figure 5.** Schematic of a simple neural network (a 'multi-layer perceptron'). The network takes an $N$-dimensional vector, $\mathbf{x}$, on the left side and transforms it into an $M$-dimensional vector $\mathbf{y}$ on the right side. Each grey box represents a 'neuron', and is a simple mathematical operation which takes many inputs (lines coming in from the left), and returns a single output value which is sent to every neuron in the next 'layer' (lines coming out from the right). The neuron's behaviour is governed by a unique set of 'weights' ($\mathbf{w}$): one common mathematical relation has a neuron in a layer with $K$ inputs return $y = \tanh(w_0 + \sum_{i=1}^{K} w_i x_i)$, where the single output element $y$ will become part of the inputs for the next layer.

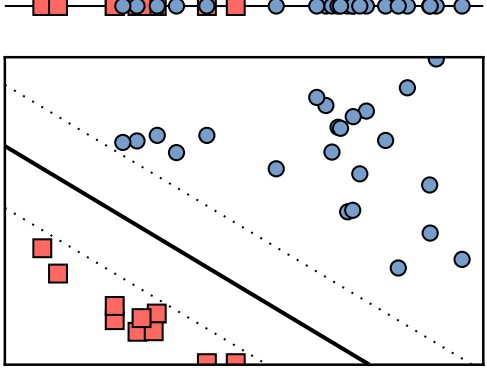

**Figure 6.** Segmenting datasets with linear decision boundaries. In the original one dimension (top), it is not possible to separate red squares from blue circles. However, by mapping this dataset into an artificial second dimension, it becomes possible to draw a linear 'decision boundary' that distinguishes the two classes. The support vector machine provides a technique for achieving this in such a way that the 'margin' between the boundary and the nearest data points is maximised (as shown by the dotted lines).

found in Bishop (1995); the aforementioned book by Mackay (2003) also discusses them at some length. Some readers may also be interested in a review by Mas and Flores (2008), targeted at the remote sensing community.

A neural network is constructed from a large number of interconnected 'neurons'. Each neuron is a processing unit that takes a number of numerical inputs, computes a weighted sum of these, and then uses this result to compute an output value. The behaviour of the neuron can therefore be controlled by altering the weights used in this computation. By connecting together many neurons, with the outputs from some being used as inputs to others, a complex system—or network—can be created, as in Fig. 5. The weights associated with each neuron are unique, so that the behaviour of the network as a whole is controlled by a large number of adjustable parameters. Typically, these are initially randomised; then, a 'training' procedure is used to iteratively update the weights until the network exhibits the desired behaviour for a given training set.

Such a brief description glosses over the richness of approaches within 'neural networks': choices must be made regarding how individual neurons behave, how they are connected together (the 'network architecture'), and how the training algorithm operates. These are generally somewhat application-dependent, and may dictate how effectively the network performs. The simplest form of neural network, sometimes called a 'multi-layer perceptron' (MLP), consists of neurons arranged in two or three 'layers', with the outputs from neurons in one layer being used as the inputs for the next layer (see Fig. 5). Traditionally, these are trained by 'back-propagation' (Rumelhart et al., 1986), which involves calculating how network outputs are influenced by each in-

dividual weight. The canonical use for such a network is as a classifier (e.g. Lippmann, 1989; Bischof et al., 1992), although they are exceptionally versatile: for example, Ermini et al. (2005) demonstrate that MLPs can be used for landslide susceptibility prediction. They may also be used to model a wide variety of physical relationships between observables, for example, in interpolation problems such as estimating difficult-to-measure surface properties from satellite observations (e.g. Krasnopolsky and Schiller, 2003), and to detect unusual or unexpected features within datasets (e.g. Markou and Singh, 2003).

In recent years, attention has increasingly focussed on 'deep learning', where many more layers of neurons are used. This has proven effective as a means to 'discover' structure in large, complex datasets, and represent these in a low-dimensional form (Hinton and Salakhutdinov, 2006). Typically, these require specialised training algorithms, since the number of free parameters in the network is exceptionally large. Such systems are particularly useful for applications in cluster analysis and dimension reduction, and these properties can be exploited to enable cataloguing of geomorphological features in large datasets, as in Valentine et al. (2013).

### 3.5 Support Vector Machines

The modern concept of the Support Vector Machine (SVM) stems from the work of Cortes and Vapnik (1995), although this builds on earlier ideas. The approach is targeted towards classification problems, framed in terms of finding, and then utilising, 'decision boundaries' that separate one class from another. In the simplest case, linear decision boundaries can be found—that is, any two classes can always be separated by a straight line when two-dimensional slices through the

dataset are plotted on a graph. The SVM method provides an algorithm for constructing linear decision boundaries that maximise the 'margin' between boundary and adjacent data points, as shown in Fig. 6.

However, in most realistic cases, the dataset cannot be cleanly categorised using linear boundaries: all possible linear decision boundaries will misclassify some data points. To handle this scenario, the SVM approach uses a mathematical trick to create nonlinear decision boundaries in a way that is computationally tractable. Data is first mapped into a higher-dimensional 'feature space' (the opposite of dimensionality reduction); in this space, the data can then be separated using linear boundaries (see Fig. 6). This mapping, or transformation, may be nonlinear, so that lines in the feature space may correspond to curves in the original data space. Various extensions to this approach exist that allow for a less-than-perfect division of the dataset, to reflect the presence of classification errors or observational noise. Once boundaries have been determined, the SVM may then be used for classification of new examples. SVMs have some similarities in structure to simple neural networks, although the 'training' or optimisation procedure is quite distinct.

Again, landslide susceptibility assessment offers one aspect of geomorphology where SVMs have found significant application (e.g. Brenning, 2005; Yao et al., 2008; Marjanović et al., 2011; Peng et al., 2014). They have also been used to differentiate between fluvial and glacial river valleys (Matías et al., 2009). Another similar use can be found in Beechie and Imaki (2014): there, the authors use an SVM to classify river channel morphologies based on geospatial data including DEMs, precipitation, and geology. This can then be used in river conservation and restoration to infer the natural patterns that existed in areas that have been subject to extensive human intervention.

## 3.6   Self-Organising Maps

The concept of the Self-Organising Map (SOM) stems from the work of Kohonen (1990), and it can be viewed as a particular class of neural network. The SOM implements a form of dimensionality reduction, and is generally used to help identify clusters and categories within a dataset. The basic premise is to take a (usually) two dimensional grid and, through training, create a mapping between that 2D space and a higher-dimensional dataset (see Fig. 7). The 2D representation can then be used to help visualise the structure of the data. The SOM is also typically designed to be significantly smaller than the training set, spanning the same data space with fewer points, and is thus easier to analyse.

To create an SOM, we start with a map consisting of a number of 'nodes', often arranged as a regular grid, so that it is possible to define a 'distance' between any two nodes, and hence identify the set of nodes that lie within a certain radius of a given node, known as its 'neighbourhood'. Each node is associated with a random 'codebook vector' with the same

dimensionality as the data (Fig. 7a). During training, we iteratively select a data example at random from the training set, identify the node with the closest-matching codebook vector, and then adjust this vector, and those of neighbouring nodes, to better match the training example (Fig. 7b). Given sufficient training, the codebook vectors come to mirror the distribution of data in the training set (Fig. 7c). Typically, the extent to which codebook vectors are updated and the radius used to define the neighbourhood of a given node are reduced as training proceeds, to promote fine-tuning of performance.

Once the SOM is trained, various approaches exist to enable visualisation of the codebook vectors, with the goal of highlighting any underlying structure within the dataset. One common approach is to try to identify clusters in the dataset by examining the distances between the codebook vectors for neighbouring nodes, often by plotting the SOM grid coloured according to these distances (sometimes described as depicting the 'U-matrix'). Alternatively, a method can be used to distort the grid in such a way that when plotted in 2D, the distance between nodes is proportional to the distance between their codebook vectors; one common technique for this is 'Sammon's mapping' (Sammon, 1969). Another visualisation approach, sometimes called 'component plane analysis', looks for correlations between input parameters, e.g. between rainfall and elevation or slope orientation and landslide risk. Here, the SOM grid is coloured according to the values of particular elements of the codebook vectors, with each element corresponding to an input parameter. Correlated parameters can then be identified by their similar colour patterns.

Potential applications in geomorphology are numerous. Marsh and Brown (2009) use an SOM-based method to analyse and classify bathymetry and backscatter data, allowing near-real-time identification of regions with particular seafloor characteristics, e.g. for benthic habitat mapping. Similarly, Ehsani and Quiel (2008) demonstrate that SOMs can be used to classify topographic features, identifying characteristic morphologies contained within DEMs such as channels, ridge crests, and valley floors. In a third example, Friedel (2011) uses SOMs to assess post-fire hazards in recently burned landscapes. Both PCA and K-means clustering are then used to divide the 540 areas studied into 8 distinct groups, which the author suggests could be used for focusing future field research and development of empirical models. As illustrated by these examples, one of the key strengths of the SOM method is that learning is unsupervised, and does not rely on the user having any prior knowledge of the 'important' structures contained within the dataset.

## 3.7   Bayesian inference

To conclude this section, we mention the concept of Bayesian inference. This is a much broader topic than any of the methods discussed so far; indeed, in many cases these methods are themselves formally derived from Bayesian concepts. Bayes' theorem (Bayes, 1763) underpins a significant frac-

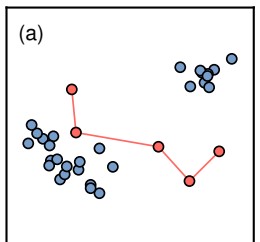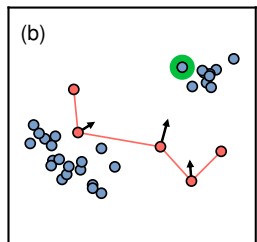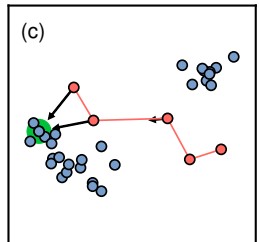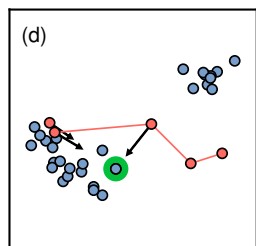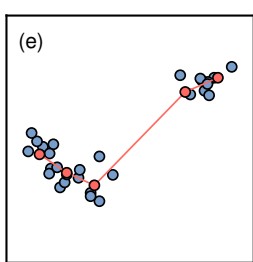

**Figure 7.** The self-organising map (SOM). A dataset consists of numerous data points (blue), and is to be represented by a one-dimensional SOM (red). The SOM contains of a number of nodes, with a well-defined spatial relationship (here, depicted by lines connecting points). Initially, all SOM nodes are associated with random locations in data space (a). During training (b, c, d) a datum is selected at random (highlighted in green). The closest SOM node is identified, and this and its nearest neighbours are adjusted so as to be closer to the chosen point (arrows), with the scale of adjustment proportional to the separation. After many iterations of this procedure, the distribution of SOM nodes mirrors that of the dataset (e). For ease of illustration, this figure depicts a two-dimensional dataset and a one-dimensional SOM; in practice, the dataset will usually have much higher dimensionality, and the SOM is usually organised as a two-dimensional grid.

tion of modern statistical techniques, and explains how new observations can be used to refine our knowledge of a system. It tells us that the probability that the system is in a certain state, given that we have made a particular observation ('obs.'), $P(\text{state}|\text{obs.})$, can be expressed in the form

$$P(\text{state}|\text{obs.}) = \frac{P(\text{obs.}|\text{state})P(\text{state})}{P(\text{obs.})}. \qquad (2)$$

Here, $P(\text{state})$ represents our prior knowledge—that is, our assessment of the probability that the system is in the given state without making any measurements—while $P(\text{obs.})$ represents the probability with which we expect to obtain the exact measurements we did, in the absence of any knowledge about the state of the system. Finally, $P(\text{obs.}|\text{state})$ expresses the probability that we would get those measurements for a system known to be in the relevant state.

In many cases, it is possible to estimate or compute the various probabilities required to implement Bayes' theorem, and thus it is possible to make probabilistic assessments. This is often useful: for example, hazard assessment is generally better framed in terms of the chance, or risk, of an event, rather than attempting to provide deterministic predictions. An extensive discussion of Bayesian analysis can be found in Mackay (2003). Sivia (1996) is another useful reference, and a paper by Griffiths (1982) provides some geomorphological context. A wide range of computational techniques have been developed to enable Bayesian calculations in various settings, for which a review by Sambridge and Mosegaard (2002) may be a good starting point.

As a simple example, suppose we are interested in classifying land use from satellite imagery. Grassland will appear as a green pixel 80% of the time, although it may also be brown: thus, $P(\text{green}|\text{grass}) = 0.8$. On the other hand, desert environments appear as brown in 99% of cases. A particular region is known to be mainly desert, with only 10% grassland—so an image of the area will consist of 8.9% green pixels $(= ((0.01 \times 0.9) + (0.8 \times 0.1)))$. If we look at one specific pixel, and observe that it is green, Bayes' theorem

tells us there is a 90% chance that the location is grassland: $P(\text{grass}|\text{green}) = P(\text{green}|\text{grass})P(\text{grass})/P(\text{green}) = 0.8 \times 0.1/0.089 \approx 0.899$. On the other hand, if a pixel is seen to be brown, we can classify it as desert with 98% certainty. This illustrates a property that emerges from the Bayesian formalism: unusual observations convey more information than routine ones. Before we obtained any satellite imagery, we could have guessed with 90% accuracy that a particular location in the region was desert; observing a brown pixel leads to only a relatively modest increase in the certainty of this determination. However, if we were to observe a green pixel, the chance that the location is a desert drops markedly, to only 10%.

Most examples in geomorphology again come from landslide susceptibility: Lee et al. (2002) and Das et al. (2012) use Bayesian techniques to assess landslide susceptibility in Korea and the Himalayas, respectively. In a similar application, Mondini et al. (2013) map and classify landslides in southern Taiwan using a Bayesian framework. Examples from other areas of geomorphology include Gutierrez et al. (2011), who use a Bayesian network to predict shoreline evolution in response to sea-level change, and Schmelter et al. (2011), who use Bayesian techniques for sediment transport modelling.

## 4  Practical considerations

Each of the techniques discussed in the previous section—and the wide variety of alternatives not mentioned here—have their strengths and weaknesses, and a number of practical issues may need to be considered when implementing a solution to a particular problem. Here, we discuss some topics that may be relevant across a range of different approaches, and which may affect the viability of a learning algorithm solution in any case.

## 4.1 Constructing a training set

Unsurprisingly, the training data used when implementing a learning algorithm can have a major impact upon results: everything the system 'knows' is derived entirely from these examples. Any biases or deficiencies within the training data will therefore manifest themselves in the performance of the trained system. This is not, in itself, necessarily problematic—indeed, the landform cataloguing system introduced by Valentine et al. (2013) relies upon this property—but it must be borne in mind when tools are used and results interpreted. If training data is low-quality or contains artefacts, glitches and other observational 'noise', results will suffer. In general, time invested in properly selecting and processing datasets will be well-spent. However, it is also important that training data remains representative of the data that will be used during operation.

One particular issue that can arise stems from the fact that the learning algorithm lacks the trained researcher's sense of context: it has no preconception that certain structures in the data are more or less significant than others. For example, suppose a system is developed to classify valley profiles as being formed by either a river or a glacier. The training data for this system would consist of a sequence of hand-classified valley profiles, each expressed as a vector of topographic measurements. If, for the sake of example, all glacial examples happen to be drawn from low-elevation regions, and all riverine examples from high-elevation regions, it is likely that the system would learn to treat elevation as an important factor during classification.

A second potential pitfall arises from the statistical basis underpinning most learning algorithms. Typically, the extent to which a particular feature or facet of the dataset will be 'learnt' depends on its prevalence within the training examples as a whole. This can make it difficult for a system to recognise and use information that occurs infrequently in the training data, since it gets 'drowned out' by more common features. Again, considering the problem of valley classification: if 99% of training examples are glacial, the system is likely to learn to disregard its inputs and simply classify everything as glacial, since this results in a very low error rate; for best results, both types should occur in roughly equal proportions within the training set. As before, this should be regarded as a natural property of the learning process, rather than as being inherently problematic; indeed, it can be exploited as a tool for 'novelty detection', allowing unusual features within a dataset to be identified (for a review, see e.g. Marsland, 2002). Nevertheless, it is a factor that ought to be borne in mind when a system is designed and used.

To avoid problems, it is important to choose training data with care, and to develop strategies for evaluating and monitoring the performance of the trained system. It is often beneficial to invest time in finding the best way to represent a given data type, so as to accentuate the features of interest, and remove irrelevant variables. This process is sometimes referred to as 'feature selection' (e.g. Guyon and Elisseeff, 2003). Thus, in the situation mentioned above, the valley profiles might be more usefully represented as variations relative to their highest (or lowest) points, rather than as a sequence of absolute elevations. In line with our comments in Section 3.2, it is often helpful to 'de-mean' and normalise the data: using the training set, it is straightforward to compute the mean input vector, and the standard deviation associated with each component of this. Then, all examples—during training, and during operation—can be standardised by subtraction of this mean, and rescaling to give unit standard deviation. Clearly, the inverse of these transformations may need to be applied before results are interpreted. For more complex applications, specialist representations such as the 'geomorphons' developed by Jasiewicz and Stepinski (2013) may be beneficial, and provide a targeted route to encoding geomorphological information. Training examples should be chosen to cover a spread of relevant cases, perhaps including different regions or measurements made at different times of year, to avoid introducing unintended biases into results.

## 4.2 Assessing performance

For 'supervised' learning algorithms, where a pre-defined relationship or structure is to be learnt, it is possible to assess performance using a second set of user-selected data—often referred to as a 'test' or 'monitoring' dataset. This test data intended to provide an independent set of examples of the phenomena of interest, allowing quantitative measures of effectiveness to be evaluated; this strategy is sometimes referred to as 'cross-validation'. It is important to do this using examples separate from the training set, to ensure that the system's 'generalisation performance' is measured: we want to be sure that the algorithm has learned properties that can be applied to new cases, rather than learning features specific to the particular examples used during training. As an analogy: a dog may learn to respond to particular commands, but only when issued in a very specific manner; it cannot then be said to properly understand a spoken language.

The metric by which performance is assessed is likely to be situation-dependent. In general, a supervised learning algorithm will be designed to optimise certain 'error measures', and these are likely to provide a good starting point. Nevertheless, other statistics may also prove useful. For classification systems, analysis of 'receiver operating characteristics' (ROCs) such as hit and false positive rates may be instructive (e.g. Fawcett, 2006). More difficult to quantify, but still potentially valuable, is the experienced researcher's sense of the plausibility of a system's predictions: do results exhibit geomorphologically reasonable patterns? For example, does a prediction of landslide risk seem plausible in its relationship with the topography and underlying bedrock?

Assessing performance in unsupervised learning is more challenging, as we may not have prior expectations against

which to measure results: fundamentally, it may be challenging to define what 'good performance' should signify. In many cases, application-specific statistics may be helpful—for example, in cluster analysis it is possible to calculate the standard deviation of each cluster, quantifying how tightly each is defined—and again, the researcher's sense of plausibility may provide some insight. It may also prove instructive to repeat training using a different subset of examples, to assess how stable results are with respect to variations in the training data: a structure or grouping that appears consistently is more likely to be real and significant than one that is very dependent on the precise examples included in the training set.

### 4.3   Overtraining and random noise

The phenomenon of 'overtraining' or 'overfitting', which typically arises in the context of supervised learning algorithms, has already been alluded to. It occurs when an iterative training procedure is carried out for too many iterations: at some point, the system usually begins to learn information that is specific to the training examples, rather than being general. This tends to reduce the performance of the system when subsequently applied to unseen data. It can often be detected by monitoring the algorithm's generalisation performance using an independent set of examples as training proceeds: this enables the training procedure to be terminated once generalisation performance begins to decrease. In certain cases, post-training strategies can be used to reduce the degree of over-fitting: the example of 'pruning' decision trees has already been mentioned. It has also been shown that 'ensemble methods' may be useful (indeed, 'random forests' provide one example of an ensemble method), whereby multiple instances of the same learning algorithm are (over-)trained, from different randomised starting points, and their outputs are then somehow averaged (e.g. Dietterich, 2000). Because each instance becomes sensitive to distinct aspects of the training data, due to their different initialisation conditions, each performs well on a subset of examples, and poorly on the remainder. However, the overall performance of the ensemble (or 'committee') as a whole is typically better than that of any individual member.

Another strategy that is adopted is to add random noise to the training data. The rationale here is that training examples typically exhibit an underlying pattern or signal of interest, overprinted by other processes and observational errors. In order to learn the structure of interest, we wish to de-sensitise our training procedure to the effects of this overprinting. If we can define a 'noise model' that approximates the statistical features of the unwanted signal, adding random realisations of this to each training example allows us to limit the extent to which the algorithm learns to rely on such features: by making their appearance random, we make them less 'useful' to the algorithm. Returning again to the example of valley classification, local variations in erosion and human

intervention might be modelled as correlated Gaussian noise on each topographic measurement. During training, each example is used multiple times, with different noise on each occasion; in theory, this results in only the gross features of the valley profile being taken into account for classification purposes. However, it may be challenging to identify and construct appropriate noise models in many realistic cases.

It is worth noting here that a similar strategy may prove useful in other cases where it is desirable to desensitise systems to particular aspects of the data. For example, spatial observations are typically reported on a grid, aligned with geographic coordinates. However, natural phenomena typically do not display any such alignment, and orientation information may be irrelevant in many cases. If 2D spatial information is used in a particular case, it may be desirable to make use of multiple, randomly-rotated copies of each training set example. This allows an effectively-larger training set to be created, and reduces the chance that features are treated differently depending on their alignment.

### 4.4   Operational considerations

As with any analysis technique, results obtained using learning algorithms can be misleading if not treated carefully. This is especially true where a technique invites treatment as a 'black box', and where the mechanism by which it operates is not always easily understood. The great strength of artificial intelligence is that it enables a computer to mimic the experienced researcher—but this is also a potential drawback, tending to distance the researcher from their data. In some sense, this is an inevitable consequence of the ever-increasing quantity of data available to researchers—but there is a risk that it leads to subtleties being missed, or results interpreted wrongly due to misapprehensions surrounding computational processing. To minimise the risk of these issues arising, it is important that researchers develop heuristics and procedures that enable 'intelligent' systems to be monitored. For example, users of automated data classification systems should monitor the statistical distributions of classification outputs, and investigate any deviations from the norm; it is also desirable to spot-check classifications. This is particularly true in settings where there is a risk that new examples lie outside the scope of the training set—perhaps where data are drawn from new geographic regions, or were collected at a different time of year.

Learning algorithms have immense potential in enabling exploration of large, complex datasets and in automating complicated tasks that would otherwise have to be done manually. However, developing a learning algorithm system—especially those targeted at classification, regression, or interpolation—can also be time-consuming and resource-intensive. Computational demands may also be significant: although many applications require no more than a standard laptop or desktop computer, some may only be viable with access to large-scale parallel computing resources. By way

of an illustration for the more computationally-intensive end of the spectrum: training the learning algorithm used to catalogue seamounts in Valentine et al. (2013) currently requires a few hundred CPU-hours on a 1.9 GHz machine, in addition to considerable computational demands for processing the raw bathymetric data. Thus, learning algorithms do not currently present an economic solution to all problems—although this balance will undoubtedly change as technology and algorithms continue to evolve.

## 5   Outlook

In this paper, we have attempted to provide a general survey of the field of learning algorithms, and how they might be useful to researchers working in the fields of geomorphometry and Earth surface dynamics. These fields benefit from extensive, large-scale, feature-rich datasets, bringing opportunities and challenges in equal measure. Although currently dominated by a few specific topics in the field, such as landslide hazard assessment, the use of artificial intelligence to help explore, process and interpret geomorphological data is almost certain to be an increasingly significant aspect of research in coming years.

An increased use of learning algorithms in geomorphological communities is likely to require developments in computational infrastructure. There are obvious requirements for access to appropriate hardware and software resources, as well as skill development. In particular, larger problems or more complex algorithms may make use of parallel computing and other high performance computing techniques essential. In addition, in light of the potentially substantial development cost of implementing some of the more complex learning algorithms, it is worth trying to plan for a flexible implementation. Most of these approaches can, in principle, be implemented in a fairly general framework, allowing the same underlying algorithm to be applied to many problems.

The computational implications of both parallelisation and generalisation are beyond the scope of this review, but one area of particular relevance to the geomorphology community concerns data input and output, and hence data format: The ability to reuse an algorithm across multiple data types is a key element of flexibility in a field with such a diversity of measurements. The ability to handle large file sizes and support efficient data access, potentially in parallel, is also an important consideration. As these techniques develop, this places increasing importance on the development and use of robust community standards for data formats. File frameworks, which allow the development of multiple specialist file formats all adhering to a common set of rules, may be particularly valuable in combining consistency from an algorithmic point of view with flexibility to accommodate varied data.

However, learning algorithms are not a panacea. 'Traditional' approaches to data analysis will remain important for the foreseeable future, and are well-suited to tackling many problems; we do not advocate their wholesale replacement. In addition, it is important that the use of more advanced computational techniques is not allowed to become a barrier between researchers and data; it is almost certain that the nature and practice of 'research' will need to evolve to accommodate increasing use of these technologies. Some interesting perspectives on these issues may be found in—for example—an issue of *Nature* dealing with '2020 Computing' (e.g. Muggleton, 2006; Szalay and Gray, 2006), and in work aimed at developing a 'robot scientist' (e.g. King et al., 2009). Nevertheless, artificial intelligence opens up many new possibilities within the fields of geomorphology and Earth surface processes—particularly given the common scenarios of large datasets; complex, interacting processes; and/or natural variability that make many geomorphological situations difficult to classify or model using conventional techniques. Learning algorithms such as those discussed here thus represent a powerful set of tools with the potential to significantly expand our capabilities across many branches of the field.

**Acknowledgements.**  We are grateful to the associate editor, John Hillier, and to Niels Anders, J.J. Becker, Ian Evans and Evan Goldstein for reviews and comments on the initial draft of this manuscript. We also thank Jeannot Trampert for numerous useful discussions. APV is supported by the European Research Council under the European Union's Seventh Framework Programme (FP/2007-2013)/ERC Grant Agreement no. 320639.

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
