# Peer review of "An introduction to learning algorithms and potential applications in geomorphometry and earth surface dynamics"

_Earth Surface Dynamics, 2016_

## Short Comment (SC1) · 25 Feb 2016

In section 4.1 the authors discuss training data. They state, on P. 17 L. 1-2:

"To avoid problems, it is important to choose training data with care, and to develop strategies for evaluating and monitoring the performance of the trained system."

My own experience with machine learning suggests that this is crucial: if the training data is bad (e.g., not enough, too much, too noisy, doesn't cover the solution space adequately, etc.), the learner cannot do its job effectively (hence the expression 'garbage in, garbage out'). My recommendation is the authors present specific guidance on how to select appropriate training data and point the reader toward such guidance in the

literature (i.e., selection routines). Note that this goes beyond offering advice on how to represent data to the learner (discussed on p. 17, line 2-8).

Respectfully,

Evan B Goldstein

Department of Geological Sciences

University of North Carolina-Chapel Hill

---

## Referee Comment (RC1) · I. Evans (Referee) · 11 Mar 2016

Comments by Dr. Ian S. Evans, Durham University, 6 March 2016 on - Earth Surf. Dynam. Discuss., doi:10.5194/esurf-2016-6, 2016

An introduction to learning algorithms and potential applications in geomorphometry and earth surface dynamics. Andrew Valentine and Lara Kalnins

GENERAL COMMENTS:

This is a general survey, essentially a tutorial, of a broad field of methods. It provides a lot of common-sense advice. It is readable (more than anticipated!), and well written except for some overuse of 'we'. I am in favour of the use of personal pronouns for

emphasis, even in scientific writing, but four 'we's in four lines (p.2, lines 17-20) seems excessive. Also 'we aim to give readers...' (p.1 line 22) comes across as condescending: I would say 'This review is intended to provide...'

Figs. 2, 3, 6 and 7 use hypothetical data sets (bivariate scatters of points). This is less persuasive than the use of real examples. If made-up data have to be used, however, could they at least be more realistic? Bivariate scatters of real data tend to fade away toward the edges, and clusters are hardly ever as separated as in Figs. 6 and 7. It should not be difficult to replace these Figs. – or even to find comparable scatters that are real observations.

Many papers use K-Means as a 'black box': Fig.2 commendably provides a good illustration of how this algorithm works. The same cannot be said, however, about Fig.7 (SOM): I do not see how (b) leads on to (c), and neither the text nor the caption enlightens me.

One reason for resorting to more complex methods is because of the non-linearity of relationships (some are triangular) and the non-sphericity of clusters in property space. This can be hinted at, even in 2-D graphs: e.g. the separating line in a redrawn Fig. 6 could be curved.

The case for black boxes or learning algorithms needs to be based on the inadequacy of simpler approaches. On p.8 (line 4), porosity is considered a function of water depth and distance. That would seem to be a case for straightforward regression with two quantitative controls, on transformed scales if necessary.

The paper provides a good read and a useful overview and juxtaposition of decision trees, K-Means, PCA, Neural networks, SVM and SOM. It covers the range of statistics from Pearson to Bayes. Although I am not about to use learning algorithms, I found a few useful references. But the case made would be much more persuasive if the paper were thoroughly revised to use real examples throughout (that in Fig. 4 is appreciated); to focus on situations where use of these methods can be shown to improve results

considerably over 'traditional' methods; and even to show where they have led to new insights into environmental relationships.

I am relieved that the authors 'do not advocate . . . wholesale replacement' of 'traditional' approaches. My approach is to keep to the simplest methods that work, and to turn 'black boxes' into transparent boxes wherever possible. If a classification produces highly overlapping classes on most dimensions, I am not interested.

DETAILS:

(page/line) 2/13 in which software context should readers 'begin by writing their own. . .'?

2/16 what are 'optimised libraries'?

2/34 'as black boxes'

3/12 the example is of 'diverse' observations, not just multiple.

3/17 the title of section 2 led me to expect specific applications. These actually come in later sections: as 2 is very general, perhaps that should be in the heading.

4/8 to 11 this is a bit too general. Surely a classification can be evaluated only when the aspects of the data on which it is based can be understood?

5/9 why do coordinates 'generally have little physical meaning'?

6/14 & 15 a desktop computer is contrasted with 'a modern machine'. This is very vague. Desktops can be modern?

Fig.1 I prefer 'standardised' to 'normalised' here: the latter may imply a normal distribution.

7/3 'the data are perfect'

9/29 I think it is the components / new variables / dimensions that are uncorrelated – not 'each parameter'.

Fig.3 shows an even distribution of data points within a sharply-bounded rectangle. I never saw an earth science data set like that: could we have a more realistic scatter?

Fig.4 is interesting: I see differences between the three rows. In the top row, N=100 captures the original thoroughly; in the second, N=200 is required: and in the bottom row only N=500 captures the lineations in the original. Thus I disagree with 'using only 100-200 dimensions', at the end of the caption.

Fig.6 Actually one dimension (y) DOES separate blue from red: they overlap only in x. This is because an unrealistically wide separation has been portrayed: again, the hypothetical example could at least approach realism.

19/6 & 7 I am unsure about 'various', but as it implies more than one, it should read 'statistical distributions of the ... classifications'.

19/16 There is no point mentioning CPU hours unless some idea of the machine involved is given.

19/26 'as well as skill'

---

## Short Comment (SC2) · 17 Mar 2016

Page 9 line 5

As the 3rd author of Martin 2015, and the main author and "debugger" of the code used there, I wanted to throw in my two cents worth about how important it is to normalize both the "features" (aka predictors) and the "response" (aka measurements).

This doesn't need to be complex, simply "Student-ize" by removing the mean of every column of the training data, (store that number), and dividing the detrended data by it's standard deviation, (also store that).

When making a prediction after training, it is important to apply the mean and std dev

of the TRAINING data to the data used in the PREDICTION.

Often "software packages" do this automatically (aka Random Forests in MATLAB), but other times this is not the case and it will dramatically hurt the prediction. For example, the recent advances in "deep learning" for Machine Vision were only achieved after the researchers understood they needed to subtract the "average image" obtained from all the training images -BEFORE- training and prediction...

-jj

---

## Referee Comment (RC2) · NS Anders (Referee) · 20 Mar 2016

This paper sums up the most widely used learning algorithms to extract information from geospatial data sources. The geospatial analysis and geomorphometry related research is becoming more and more data-driven and I encourage papers advocating making use of modern computational techniques in geomorphology. This paper is particularly interesting to those unfamiliar to learning algorithms. Below I have addressed some comments that could improve the paper.

General

* The paper explains well the most widely used methods in current literature with effec-

tive illustrations. Each topic is supported with one or few references to applications in the geosciences, mostly landslide susceptibility. However, to justify the part in the title concerning '...and potential applications in geomorphometry and earth science dynamics' I think the paper should cover a more in depth section on the potential applications of learning applications in the geosciences. Or perhaps remove 'potential' from the title and at least provide more references in the 3.1-3.7 subsections to current research that are applying these algorithms.

* Also in the subsections there are occasional statements such as 'it is claimed that' (P4L8) or 'literature describes a rule of thumb' (P6L9), which are not followed by references. I think it is required to provide references to support such statements in general.

* I suggest presenting an overview of strengths and weaknesses in different scenarios (data sets), or what learning algorithm could be appropriate for particular data sets or different scenarios, possibly as a table. For example when should scientists choose random forests over neural networks. There is probably not a single best method for each scenario, but I guess some algorithms are ruled out because of certain data types or data properties (maybe due to statistical limitations). This would also broaden the audience reading this paper.

* Although the English is much better than my own, there are a few singular/plural errors throughout the paper. I suggest to have the paper thoroughly checked.

Specific

P3L11: I would not say LiDAR 'image'. In remote sensing an image is often associated with spectral measurements captured by a sensor creating this image. The authors probably mean a derived grid of interpolated elevation values from a LiDAR point dataset, hence LiDAR DEM or DTM is more appropriate than 'image'. Also the abbreviation style LiDAR (with capital A) is more commonly used than LiDaR.

P4L33: It is unclear to me what exactly this 'data vector' means with respect to a grid

and why a 100m x 100m grid with with 10 m cells (thus 10x10=100 cells) would have (11x11=) 121 dimensions.

P5L6-10: I wonder if dimension reduction by means of visualization is appropriate to mention in this section, as dimension reduction in the context of learning algorithms is primarily used to reduce complexity of calculations and/or computing. To me the way how results are visualized and whether dimensions or detail are lost is a different discussion.

P5L11-15: While vegetation of course has indirectly also geomorphological implications I'd like to, rather shamelessly, self-advertise our recent paper that is a more pure example of cluster analysis in (palaeo-glacial) geomorphology, see Smith et al (2016) in ESPL: http://onlinelibrary.wiley.com/doi/10.1002/esp.3828/abstract

P6L10: is 'performance' here meant in terms of computation time, or in terms of accuracy, or both.

P9L18: now LiDAR is abbreviated as LIDAR. I suggest selecting one style of abbreviation, preferably Lidar, LiDAR or LIDAR.

Section 3.3. I think PCA should be explained in more detail. It is for example not mentioned what the axes actually mean and that what information is derived from the point distribution along the axes. For example that the axes do likely resemble an average of multiple dimensions is not mentioned. For example in the example of a PCA of plant species distribution with respect abiotic factors such as soil type, ph, and water availability a single PCA axis could resemble both ph and water availability and could make interpretation of PCA more complex. When describing PCA I think such information is essential to mention.

---

## Editor Comment (EC1) · J. K. Hillier (Editor) · 31 Mar 2016

Dear Drs Valentine and Kalins,

The comments and review of your manuscript appear detailed, generally supportive, and constructive. So, I very much look forward to seeing a revised manuscript.

John
* * *

---

## Author Comment (AC1) · 28 Apr 2016

Department of Earth Sciences
Universiteit Utrecht

April 28, 2016

Dr J.K. Hillier
Associate Editor, ESurf

Dear Dr Hillier,

We enclose an updated version of our manuscript, 'An introduction to learning algorithms and potential applications in geomorphometry and earth surface dynamics'. We are grateful to the reviewers and commenters for their constructive criticisms on the original draft. Our responses to their substantive points are set out below. A copy of the manuscript marked up to highlight the changes we have made is appended to this letter.

**RC1 (Ian Evans)**

> This is a general survey, essentially a tutorial, of a broad field of methods. It provides a lot of common-sense advice. It is readable (more than anticipated!), and well written except for some overuse of 'we'. I am in favour of the use of personal pronouns for emphasis, even in scientific writing, but four 'we's in four lines (p.2, lines 17-20) seems excessive. Also 'we aim to give readers. . . ' (p.1 line 22) comes across as condescending: I would say 'This review is intended to provide. . .

We have replaced the phrase on p. 1, line 22 and reduced the use of 'we' overall.

> Figs. 2, 3, 6 and 7 use hypothetical data sets (bivariate scatters of points). This is less persuasive than the use of real examples. If made-up data have to be used, however, could they at least be more realistic? Bivariate scatters of real data tend to fade away toward the edges, and clusters are hardly ever as separated as in Figs. 6 and 7. It should not be difficult to replace these Figs. – or even to find comparable scatters that are real observations.

We agree with the reviewer that the use of 'real' data sets could be regarded as more persuasive or impressive. However, this has to be offset against the complications that inevitably accompany such examples. Realistic datasets tend to be high-dimensional, making it difficult to illustrate concepts clearly using only one or two two-dimensional figures. In addition, a significant amount of explanation may be required to make examples meaningful to the full spectrum of potential readers. This diverts readers' attention from the main focus of the paper, and may introduce unnecessary sources of confusion. On balance, we are not convinced that the benefits of using more real examples outweighs the potential costs. In particular, we feel that simple, synthetic examples are generally necessary to illustrate how a particular method works; real examples would have to be in addition to, rather than instead of, these figures.

However, we agree that some of the original figures exhibited unnecessarily unrealistic features, and we have attempted to improve this in the current version.

> Many papers use K-Means as a 'black box': Fig.2 commendably provides a good illustration of how this algorithm works. The same cannot be said, however, about Fig.7 (SOM): I do not see how (b) leads on to (c), and neither the text nor the caption enlightens me.

We have altered this figure and its caption, and we hope the result is clearer.

One reason for resorting to more complex methods is because of the non-linearity of relationships (some are triangular) and the non-sphericity of clusters in property space. This can be hinted at, even in 2-D graphs: e.g. the separating line in a redrawn Fig. 6 could be curved.

In designing the figures for this paper, we have attempted to strike a balance between simplicity and realism. While the relationships used for illustrative purposes could be made more complex, it is not clear to us that this would necessarily enhance readers' understanding.

In the specific case of Fig. 6, the 'support vector machine' method is based around construction of linear boundaries separating clusters. Thus, the separating line cannot be drawn as a curve. We have modified the figure and caption in an effort to make it clearer.

The case for black boxes or learning algorithms needs to be based on the inadequacy of simpler approaches. On p.8 (line 4), porosity is considered a function of water depth and distance. That would seem to be a case for straightforward regression with two quantitative controls, on transformed scales if necessary.

The cited example (Martin et. al., 2015) makes use of a range of different datasets when generating porosity estimates, and not just the two observables we list for illustrative purposes. We have rephrased this passage in an attempt to make this clearer.

The paper provides a good read and a useful overview and juxtaposition of decision trees, K-Means, PCA, Neural networks, SVM and SOM. It covers the range of statistics from Pearson to Bayes. Although I am not about to use learning algorithms, I found a few useful references. But the case made would be much more persuasive if the paper were thoroughly revised to use real examples throughout (that in Fig. 4 is appreciated); to focus on situations where use of these methods can be shown to improve results considerably over 'traditional' methods; and even to show where they have led to new insights into environmental relationships.

I am relieved that the authors 'do not advocate...wholesale replacement' of 'traditional' approaches. My approach is to keep to the simplest methods that work, and to turn 'black boxes' into transparent boxes wherever possible. If a classification produces highly overlapping classes on most dimensions, I am not interested.

As already discussed, we are sympathetic to the call for greater use of 'real-world' examples. However—somewhat by definition—it is difficult to find problems which are straightforward to explain and illustrate, yet which cannot be convincingly tackled via 'traditional' methods. The rationale underpinning this paper is that learning algorithms are currently relatively unknown in geomorphology: thus, there are relatively few examples in the literature to draw on, and most of these are concentrated within one or two sub-fields. One option would be to make use of examples from fields other than geomorphology; however, this may introduce additional sources of confusion. We have tried to balance these competing factors by using generic examples to illustrate how the techniques work whilst increasing the number and range of references given at the end of the various subsections to provide real-world examples.

DETAILS:
(page/line) 2/13 in which software context should readers 'begin by writing their own...'?
2/16 what are 'optimised libraries'?

We have modified this passage.

2/34 'as black boxes'

Corrected.

> 3/12 the example is of 'diverse' observations, not just multiple.

Changed to 'multiple, diverse'.

> 3/17 the title of section 2 led me to expect specific applications. These actually come in later sections: as 2 is very general, perhaps that should be in the heading.

We have re-titled this section.

> 4/8 to 11 this is a bit too general. Surely a classification can be evaluated only when the aspects of the data on which it is based can be understood?

Unfortunately, we are not sure we understand the reviewer's point in the context of the cited passage. A learning algorithm classification scheme inherently uses the input data in an unprescribed way, but the performance can still be evaluated by comparison to manual classification. However, we have made some alterations to our text in an effort to make it clearer.

> 5/9 why do coordinates 'generally have little physical meaning'?

We have attempted to clarify this point.

> 6/14 & 15 a desktop computer is contrasted with 'a modern machine'. This is very vague. Desktops can be modern?

We have altered this to refer to a 'modern high-performance machine'.

> Fig.1 I prefer 'standardised' to 'normalised' here: the latter may imply a normal distribution.

Our use of the word 'normalised' is in accordance with its (mathematical) definition.

> 7/3 'the data are perfect'

Corrected.

> 9/29 I think it is the components / new variables / dimensions that are uncorrelated – not 'each parameter'.

Unfortunately, we do not understand the distinction that the reviewer is attempting to draw here. We have changed the text to refer to 'parameter or component', in the hope that this resolves any confusion.

> Fig.3 shows an even distribution of data points within a sharply-bounded rectangle. I never saw an earth science data set like that: could we have a more realistic scatter?

We have updated the figure appropriately.

> Fig.4 is interesting: I see differences between the three rows. In the top row, N=100 captures the original thoroughly; in the second, N=200 is required: and in the bottom row only N=500 captures the lineations in the original. Thus I disagree with 'using only 100-200 dimensions', at the end of the caption.

Obviously, this ultimately depends on how accurately one wishes to represent the original data, and which features are regarded as 'significant'. This may well vary according to application. We have updated the caption to reflect this.

> Fig.6 Actually one dimension (y) DOES separate blue from red: they overlap only in x. This is because an unrealistically wide separation has been portrayed: again, the hypothetical example could at least approach realism.

We have updated the figure and caption.

> 19/6 & 7 I am unsure about 'various', but as it implies more than one, it should read 'statistical distributions of the ... classifications'.

We have re-written this sentence.

> 19/16 There is no point mentioning CPU hours unless some idea of the machine involved is given.

We now state the clock speed.

> 19/26 'as well as skill'

Corrected.

**RC2 (Niels Anders)**

> This paper sums up the most widely used learning algorithms to extract information from geospatial data sources. The geospatial analysis and geomorphometry related research is becoming more and more data-driven and I encourage papers advocating making use of modern computational techniques in geomorphology. This paper is particularly interesting to those unfamiliar to learning algorithms. Below I have addressed some comments that could improve the paper.

> General
> The paper explains well the most widely used methods in current literature with effective illustrations. Each topic is supported with one or few references to applications in the geosciences, mostly landslide susceptibility. However, to justify the part in the title concerning '...and potential applications in geomorphometry and earth science dynamics' I think the paper should cover a more in depth section on the potential applications of learning applications in the geosciences. Or perhaps remove 'potential' from the title and at least provide more references in the 3.1-3.7 subsections to current research that are applying these algorithms.

We have increased the number of examples of current research, ensuring that at least three examples are given for each technique in subsections 3.1–3.7. We have also tried to ensure that a range of examples are given where possible. However, these techniques have been much more widely used in the subfield of landslide susceptibility, so studies on this topic do form a large part of the existing literature on applications of machine learning to geomorphology. One potential, if coincidental, benefit of this is that it demonstrates that a range of machine learning techniques can be successfully applied to very similar problems — there is not a single correct approach for a given problem.

Throughout the paper, we have tried to emphasise the general types of geomorphological problem or task that these techniques may be useful for, to highlight the wide range of potential applications — Section 2, in particular, focuses on this. The examples given in both Section 2 and Section 3 are also presented more to illustrate the range of what is possible than to attempt to explain in any detail what the previous authors have done or the conclusions they reached. We thus feel that 'potential applications' is a reasonable title.

> Also in the subsections there are occasional statements such as 'it is claimed that' (P4L8)

> or 'literature describes a rule of thumb' (P6L9), which are not followed by references. I think it is required to provide references to support such statements in general.

In both of these cases, we are expressing some disagreement with statements or misconceptions that—in our experience—occur quite frequently within the literature, usually introduced as 'received wisdom' without any attempt at justification. In such circumstances, we feel it is unfair to identify one or two randomly-selected papers as apparent targets for our criticism.

> I suggest presenting an overview of strengths and weaknesses in different scenarios (data sets), or what learning algorithm could be appropriate for particular data sets or different scenarios, possibly as a table. For example when should scientists choose random forests over neural networks. There is probably not a single best method for each scenario, but I guess some algorithms are ruled out because of certain data types or data properties (maybe due to statistical limitations). This would also broaden the audience reading this paper.

We have added a table that summarises the various methods discussed in the paper. This does not go as far (perhaps) as the reviewer hoped: it is not possible to enumerate all the considerations that apply when deciding which method(s) should be adopted in a given situation. Nevertheless, we hope the addition will assist readers.

> Although the English is much better than my own, there are a few singular/plural errors throughout the paper. I suggest to have the paper thoroughly checked.

We have eliminated a few typos and editing errors.

> Specific P3L11: I would not say LiDAR 'image'. In remote sensing an image is often associated with spectral measurements captured by a sensor creating this image. The authors probably mean a derived grid of interpolated elevation values from a LiDAR point dataset, hence LiDAR DEM or DTM is more appropriate than 'image'. Also the abbreviation style LiDAR (with capital A) is more commonly used than LiDaR.

We have corrected this, and ensured all abbreviations are set as 'LiDAR'.

> P4L33: It is unclear to me what exactly this 'data vector' means with respect to a grid and why a 100m x 100m grid with with 10 m cells (thus 10x10=100 cells) would have (11x11=) 121 dimensions.

There are two possible interpretations of 'a $100\,\mathrm{m} \times 100\,\mathrm{m}$ grid... sampled at $10\,\mathrm{m}$ intervals'. One—as assumed by the reviewer—involves dividing the region into 100 equal cells, and sampling at the centre of each cell, leading to 100 measurements. The second involves sampling at the corners of each cell, resulting in 121 measurements. (This is often described as a 'fence-post problem': a ten-panel garden fence requires eleven fence-posts.)

We have edited this passage in an effort to reduce the potential for confusion, and make the link to 'data vectors' more explicit.

> P5L6-10: I wonder if dimension reduction by means of visualization is appropriate to mention in this section, as dimension reduction in the context of learning algorithms is primarily used to reduce complexity of calculations and/or computing. To me the way how results are visualized and whether dimensions or detail are lost is a different discussion.

We feel that the use of learning algorithms to assist in dataset visualisation is an topic worth mentioning here. Certainly, it is an approach that is used more in some fields than in others, but we believe it is important to alert readers to this possibility. We wonder if the reviewer slightly misunderstood our

intentions here, and we have rephrased it to make things clearer.

> P5L11-15: While vegetation of course has indirectly also geomorphological implications I'd like to, rather shamelessly, self-advertise our recent paper that is a more pure example of cluster analysis in (palaeo-glacial) geomorphology, see Smith et al (2016) in ESPL: http://onlinelibrary.wiley.com/doi/10.1002/esp.3828/abstract

We have adopted this suggestion.

> P6L10: is 'performance' here meant in terms of computation time, or in terms of accuacy, or both.

We intended this to refer mainly to 'accuracy'. We have edited the text to make this clearer.

> P9L18: now LiDAR is abbreviated as LIDAR. I suggest selecting one style of abbreviation, preferably Lidar, LiDAR or LIDAR.

We have now converged on 'LiDAR'.

> Section 3.3. I think PCA should be explained in more detail. It is for example not mentioned what the axes actually mean and that what information is derived from the point distribution along the axes. For example that the axes do likely resemble an average of multiple dimensions is not mentioned. For example in the example of a PCA of plant species distribution with respect abiotic factors such as soil type, ph, and water availability a single PCA axis could resemble both ph and water availability and could make interpretation of PCA more complex. When describing PCA I think such information is essential to mention.

We have updated our discussion in the light of these comments, and added an example where the components relate to multiple observables in this way.

**SC1 (Evan Goldstein)**

> In section 4.1 the authors discuss training data. They state, on P. 17 L. 1-2: "To avoid problems, it is important to choose training data with care, and to develop strategies for evaluating and monitoring the performance of the trained system." My own experience with machine learning suggests that this is crucial: if the training data is bad (e.g., not enough, too much, too noisy, doesn't cover the solution space adequately, etc.), the learner cannot do its job effectively (hence the expression 'garbage in, garbage out'). My recommendation is the authors present specific guidance on how to select appropriate training data and point the reader toward such guidance in the literature (i.e., selection routines). Note that this goes beyond offering advice on how to represent data to the learner (discussed on p. 17, line 2-8).

We have tried to incorporate this point into our discussion in Section 4.1. However, we do not feel there is much 'specific guidance' that can usefully be given in the present paper: anything beyond the most general remarks must surely depend upon the data and application being considered.

**SC2 (J.J. Becker)**

> *Page 9 line 5*: As the 3rd author of Martin 2015, and the main author and "debugger" of the code used there, I wanted to throw in my two cents worth about how important it is to normalize both the "features" (aka predictors) and the "response" (aka measurements). This doesn't need to be complex, simply "Student-ize" by removing the mean of every column of the training data, (store that number), and dividing the detrended data by its standard deviation, (also store that). When making a prediction after training, it is important to apply the mean and std dev of the TRAINING data to the data used in the

PREDICTION. Often "software packages" do this automatically (aka Random Forests in MATLAB), but other times this is not the case and it will dramatically hurt the prediction. For example, the recent advances in "deep learning" for Machine Vision were only achieved after the researchers understood they needed to subtract the "average image" obtained from all the training images -BEFORE- training and prediction...

We have now included this point in Section 4.1.

Yours sincerely,

Andrew Valentine

[revised manuscript text omitted]